# Enhancing fibre-optic distributed acoustic sensing capabilities with blind near-field array signal processing

Felipe Muñoz[1] & Marcelo A. Soto [1✉]

Distributed acoustic sensors (DAS) can monitor mechanical vibrations along thousands independent locations using an optical fibre. The measured acoustic waveform highly varies along the sensing fibre due to the intrinsic uneven DAS longitudinal response and distortions originated during mechanical wave propagation. Here, we propose a fully blind method based on near-field acoustic array processing that considers the nonuniform response of DAS channels and can be used with any optical fibre positioning geometry having angular diversity. With no source and fibre location information, the method can reduce signal distortions and provide relevant signal-to-noise ratio enhancement through sparse beamforming spatial filtering. The method also allows the localisation of the two-dimensional spatial coordinates of acoustic sources, requiring no specific fibre installation design. The method offers distributed analysis capabilities of the entire acoustic field outside the sensing fibre, enabling DAS systems to characterise vibration sources placed in areas far from the optical fibre.

[1] Department of Electronics Engineering, Universidad Técnica Federico Santa María, 2390123 Valparaíso, Chile. ✉email: marcelo.sotoh@usm.cl

Distributed optical fibre sensors[1] have taken a great deal of attention in several application fields, due to their unique capability to perform spatially resolved monitoring of environmental quantities such as temperature, strain, pressure, humidity and so on. Exploiting the natural scattering processes originated in optical fibres[2,3], different types of distributed fibre sensors have been developed based on Raman[4–6], Brillouin[7–10] or Rayleigh scattering[11–14]. Among all these technologies, the high sensitivity of the optical phase of coherent Rayleigh scattering to external perturbations, such as vibrations, allows for the development of distributed acoustic sensors (DAS)[1,14], which are capable of monitoring vibrating mechanical waves (acoustic waves) along an optical fibre. Note however that DAS sensors only measure longitudinal strain along the sensing fibre, resulting in a directional strain response. The fast dynamic response of DAS systems, enabling wide acoustic bandwidth, and their capability to monitor the magnitude, frequency and phase of mechanical perturbations have found a wide range of applications[1,14].

High-performance DAS systems can monitor thousands of independent spatial points (so-called acoustic channels) over the sensing fibre, trading off parameters like acoustic bandwidth, fibre length, spatial resolution (commonly called gauge length) and strain resolution[1,14]. It is important to notice that the DAS acoustic response is not uniform along the sensing fibre due to the combination of different possible reasons[15–17]: (i) the acoustic signal can arrive to local fibre sections with different relative angles, inducing different or even null levels of local longitudinal strain depending on the optical fibre orientation, (ii) the strain coupling efficiency from the propagation media could vary along the optical fibre, inducing deficient local strain transfer to some acoustic channels, or (iii) the Rayleigh intensity fading affecting most of DAS systems lead to measurements with blind fibre positions, where no reliable acoustic signal can be retrieved. In addition to these intrinsic limitations, the propagating media in some scenarios can induce multiple reflections and reverberations of the mechanical wave, distorting the waveforms measured by DAS systems[18,19]. For many applications, obtaining an undistorted representation of the emitted acoustic waveform is of crucial relevance. To improve the measured signal quality, reverberations and interferences could be mitigated using dedicated acoustic processing[19–21]. In particular, the distributed feature of DAS measurements makes possible the use of array signal processing[22–24] to enhance the performance and capabilities of DAS systems. Adjusting the delay of each DAS channel along the fibre, beamforming techniques can be exploited to enhance the acoustic signal quality through spatial filtering and to localise the acoustic source.

The use of sub-array signal processing applied to DAS measurements was first demonstrated by Ku et al.[25]. In that early work, the localisation of human footsteps at 10 m distance from the sensing fibre was performed using small linear fibre sections, whilst no further details on the processing method and approach are provided. On the other hand, some works in the field of DAS-based seismology have reported the use of seismic array processing assuming a plane mechanical wave propagation[26–29], limiting processing methods to detect only the direction of arrival of seismic waves using far-field narrowband approaches. Some of these techniques make use of acoustic wave information[28] or hybrid systems based on a combination of DAS and traditional seismometers[29]. In this context, multiple signal characterisation (MUSIC)[30,31] is one of the most used array processing methods to detect the direction of arrival of acoustic waves[28], being especially suitable for narrowband signals[31]. Using specific acoustic sensing arrangements, like two or more parallel straight arrays, MUSIC can be employed to estimate different relative angles of arrival and localise the source position by intersecting the different estimations. Unfortunately, this requires a specific design of the optical fibre installation, which must follow identified straight lines separated by a well-defined distance, which depends on the spatial sampling and target acoustic frequency[32]. The localisation and tracking of moving single-tone acoustic sources have also been demonstrated based on Doppler effect by using lumped Rayleigh reflectors interrogated by an optical frequency-domain reflectometer[33]. The capabilities of acoustic array processing[22–24] to estimate the spatial coordinates (i.e., not only the angle of arrival) of more general broadband acoustic sources, and to implement a spatial filter that enhances the measured waveform quality, while dealing with the distinct local orientations of an installed optical fibre, have not been yet explored in the literature. These tasks indeed become very challenging in real-field DAS scenarios, where besides the uneven sensitivity of DAS acoustic channels and directivity of the distributed sensor, the presence of multiple reflections and reverberations can significantly impair the performance of classical array processing[22–24].

In this work, we propose a technique to enhance the capabilities of DAS systems for general broadband acoustic signals based on near-field array signal processing, which can deal with the uneven longitudinal response of DAS acoustic channels and the different local orientations of the sensing optical fibre. The method is based on a fully blind approach that evaluates and ranks the measurement reliability of each DAS channel according to the peak-to-root-mean-square ratio of the phase cross-correlation function between channels, enabling the use of non-uniform (sparse) acoustic array processing approaches. This is the first demonstration of this kind of beamforming spatial filtering approach applied to DAS technology to improve the measured acoustic waveform quality by reducing acoustic interferences caused by reflections and reverberations originated during multipath mechanical wave propagation, while also providing a relevant signal-to-noise ratio enhancement. In addition, the proposed near-field approach allows the fully blind estimation of the two-dimensional (2D) spatial coordinates of broadband acoustic sources, considering the uneven longitudinal response of DAS sensors and using no specific designs in the geometry of the optical fibre positioning, provided there is good angular diversity of the installed fibre. Since the nonuniform response of the acoustic channels partially depends on multipath mechanical wave propagation features, the performance of the proposed techniques is evaluated through a statistical analysis using several acoustic source positions. Results demonstrate that more than 50% of the analysed cases show an acoustic signal-to-noise (SNR) enhancement between 4.36 dB and 18.54 dB, while the acoustic source position can be estimated with very low relative error when compared to the actual distance between the source and optical fibre.

## Results

**Data and experimental conditions.** The DAS measurements utilised in this work correspond to a seismic survey conducted during the Poroelastic Tomography (PoroTomo) project at a geothermal site near Brady Hot Springs, Nevada, USA[34,35]. The sensing optical fibre is 8.63 km long and is installed horizontally in zigzag on the ground surface, as shown in Fig. 1a. DAS measurements are obtained with a sampling rate of 1 kHz, a gauge length of 10 m, and a spatial sampling interval of 1 m, resulting in 8630 acoustic channels along the sensing fibre.

Each dataset corresponds to DAS measurements of the 20 s chirped acoustic signal shown in Fig. 1b, emitted by a vibroseis truck and propagating from 50 different independent locations

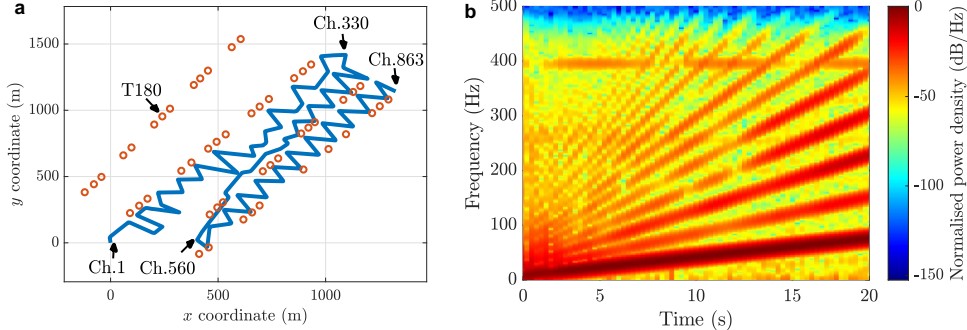

**Fig. 1 Distributed acoustic sensor setup and emitted signal. a** Sensing optical fibre (blue line) and all the acoustic source locations analysed in this work (orange circles). The sensing fibre covers an area of 0.5 × 1.5 km and is buried in a trench of 100 cm deep. The location denoted as T180 corresponds to an example position used to explain the blind techniques applied through this document. All the rest of source locations are used to perform a statistical analysis and obtain general conclusions. Some channels (e.g., Ch.1 and Ch. 863) are labelled for visual reference. **b** Spectrogram of the 20 s emitted acoustic signal, represented by a linear frequency sweep in the range from 5 to 80 Hz, with a temporal amplitude taper at the beginning and at the end of the signal. Some spectral harmonics are observed at locations near the emission point.

(orange circles in Fig. 1a) toward the sensing optical fibre. The source location T180 is here used throughout this manuscript to demonstrate the proposed blind array strategies, which is then followed by a statistical analysis based on the 50 source positions. To analyse longitudinally independent acoustic measurements, only channels separated by 10 m (equal to the gauge length) are here considered. To give homogeneity to the different datasets, margins of 5 s are left before and after the vibration waveform. In this way, each dataset comprises 863 independent acoustic channels with measurements of 30 s. Note that besides the chirped acoustic signal, some DAS channels are affected by a low-frequency interference originated by external neighbouring sources, as shown in Supplementary Fig. 2.

**Blind pilot trace selection**. An essential step required for acoustic beamforming[22,23] consists in the synchronisation of measurements obtained by all DAS acoustic channels, which are time shifted to align the signal components coming from a given analysed position. This corresponds to a spatial filtering process that requires the estimation of the time difference of arrival (TDOA)[36] among different acoustic channels, allowing the reconstruction of the acoustic waveform emitted at a specific location, which can be outside the area covered by the sensing optical fibre. Note that, even in an anisotropic propagation scenario, the estimated TDOAs can allow for the proper synchronisation of different phase-shifted DAS channels for signal enhancement, leading to a better representation of the emitted acoustic wave with a given reference phase condition. In addition, TDOA estimations can also be used in geometric positioning to convert them into distance estimations and triangulate the source position[22,23]. However, this TDOA-based source localisation method would only be effective in isotropic propagation conditions. This TDOA estimation could be quite trivial if we know the acoustic source location, the sensing fibre position, and the propagation velocity of the mechanical wave. If part of this information is not available, but we know the emitted acoustic waveform, TDOAs could still be estimated by cross correlating this reference waveform with each acoustic channel measurement and analysing the time lag of the main correlation peak[36]. Unfortunately, in the most general case of DAS monitoring, the acoustic source position, the precise distributed location of the optical fibre, the acoustic waveform, and wave propagation information are unknown, making necessary a fully blind procedure to estimate the TDOAs for each DAS channel. Here a blind TDOA estimation procedure is proposed based on the selection of a pilot trace that is used as a reference to obtain the relative time delays of all DAS acoustic channels.

A fully blind approach for DAS array processing must consider the nonuniform sensitivity of DAS channels (illustrated in Supplementary Fig. 1) and the fact that the acoustic wave arriving at each optical fibre position might propagate in an anisotropic and inhomogeneous media[14–17]. This means that the measured distributed acoustic data might contain channels that capture the acoustic signal with no major distortions, while others receive versions altered by reverberation and echoes. Therefore, compared to classical TDOA estimation[36] and array processing methods[22–24], where all acoustic measurements have similar sensitivity and are all used in the process, in DAS scenarios only some acoustic channels are useful to obtain reliable TDOA estimations. This leads to a sparse array with DAS channels nonuniformly distributed in space. This is because channels with poor acoustic sensitivity and/or containing distorted signals will mostly impair TDOA estimations instead of helping in increasing their reliability, and therefore the beamforming spatial filtering and the source location estimation will be impaired. It is then expected that, for a given number of acoustic channels and due to DAS measurement impairments, array signal processing applied to real-field DAS systems would have a lower performance compared to classical approaches using uniform sensor responses.

The method here proposed considers the selection of a pilot trace with the lowest possible level of distortion among all the measured DAS acoustic signals. To blindly find this pilot trace, we use the procedure illustrated in Fig. 2a, based on the calculation of the root-mean-square (RMS) reliability indicator $\beta$ of each DAS channel, as described in the "Methods", and selecting the one with the highest indicator. Note that this indicator is based on the calculation of the sharpness of the phase cross-correlation function (PCCF)[37] among different channels, thus being an amplitude-unbiased method to measure the similarity and phase coherence between two waveforms[37–39]. This helps us to find the TDOA that maximises the coherence between the channels, instead of considering only the more energetic components of the signal. Supplementary Fig. 4 compares the use of amplitude and phase cross-correlation functions for TDOA estimation, illustrating the advantages of phase correlation in this context. Figure 2b shows the value of the indicator $\beta_i$ as a function of the candidate channel $i$, highlighting the value of the selected pilot trace $i_p$ (channel 670 in this case). The figure clearly illustrates the variability of the indicator $\beta$ due to the uneven response of the DAS system, indicating that the measurements of some channels are more reliable as pilot traces.

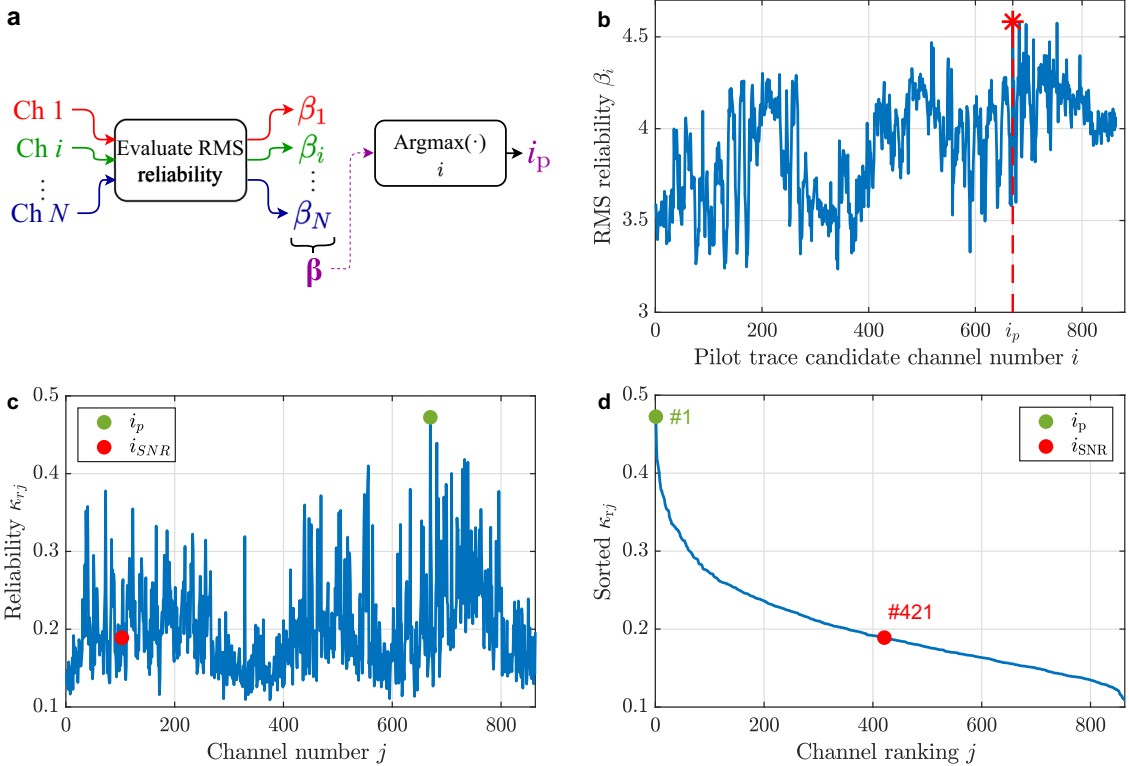

**Fig. 2 Blind pilot trace selection from data. a** The scheme shows the procedure utilised to find the most reliable channel as a pilot trace. In the process, each acoustic channel $i$ is considered as a pilot trace candidate and is compared with all the other DAS channels $j$, with $j = 1,...,N$, where $N$ is the total number of longitudinal acoustic channels. The comparison is made by calculating the phase cross-correlation function $PCCF_{ij}$ between the pair of channels $(i,j)$ and characterising the sharpness of the correlation peak using the indicator $\kappa_{ij}$ defined in the "Methods". In this way, a 1x$N$ vector defined as $\boldsymbol{\kappa}_i = [\kappa_{i,1}, \kappa_{i,2}, ..., \kappa_{i,N}]$ is obtained for the $i^{th}$ candidate. To blindly estimate the reliability of each candidate pilot trace $i$, the root-mean-square (RMS) value of the $\boldsymbol{\kappa}_i$ vector is calculated and defined as $\beta_i$. Repeating this process for $i = 1,...,N$, we obtain a 1x$N$ vector $\boldsymbol{\beta} = [\beta_1, \beta_2, ... , \beta_N]$, which contains the indicators for all the pilot trace candidates. The pilot trace channel $i_p$ is then chosen based on the channel having the highest $\beta_i$ value, since it represents the acoustic channel with the highest RMS reliability. **b** Blind evaluation of the RMS reliability indicator $\beta_i$ for all channels. **c** Non-blind validation of the method, showing the reliability indicator $\kappa_{rj}$ for all channels $j$ compared to the known emitted acoustic signal r (reference waveform). The reliability obtained by the proposed method (green circle) is compared to the one based on the use of the channel with the best signal-to-noise ratio (SNR) (red circle). **d** Reliability ranking obtained by sorting the indicator $\kappa_{rj}$. The 1$^{st}$ position in the ranking represents the channel with the highest $\kappa_{rj}$, and therefore, it is the channel with the highest similarity and phase coherence with respect to the reference signal r, while the channel at the 863$^{rd}$ position corresponds to the channel with the worst $\kappa_{rj}$.

To check the effectiveness of the proposed method for blind pilot trace selection, a non-blind validation is performed calculating the $PCCF_{ij}$ between the known emitted acoustic waveform (shown in Fig. 1b), used as a reference signal r, and each acoustic channel $j$, and evaluating the similarity indicator $\kappa_{rj}$ (see "Methods" for details). Figure 2c shows this indicator $\kappa_{rj}$ for all DAS channels, verifying that the blindly chosen pilot trace corresponds to the one with the highest similarity. For comparison purposes, the figure also shows the channel having the highest acoustic SNR measurement ($i_{SNR}$), which according to the literature can also be used as a pilot trace[40]. The result however points out that selecting the pilot trace based on the SNR is not always a good choice due to the low phase coherence that this signal may have with respect to the reference trace. This can be explained by the presence of multiple reflections originated in the analysed complex acoustic propagation scenario, which increase the SNR of a given acoustic channel by increasing the signal strength at the cost of distorting the measured waveform and reducing its similarity with the reference. Note that the same phenomenon can also occur if there exist more than one acoustic source in the analysed area. To assess the performance of the proposed method, a ranking is created by ordering $\kappa_{rj}$ from the highest to the lowest value, as shown in Fig. 2d. In this way, it is

possible to directly visualise the position of the chosen pilot channel in the ranking. The figure verifies that, for the dataset used as an example, the proposed method blindly selects the channel with the highest similarity with respect to the reference, while selecting the channel based on the SNR is not a good approach due to its low similarity have with respect the emitted waveform.

**Blind TDOA estimation.** The TDOA $\tau_{i_p,j}$ of each channel $j$ with respect to the selected pilot trace $i_p$ is obtained by finding the lag at which the absolute maximum of the $|PCCF_{i_p,j}|$ occurs[37,38] (see "Methods"). We define $\boldsymbol{\tau} = [\tau_{i_p,1}, ... , \tau_{i_p,N}]$ as a 1x$N$ vector containing all the estimated TDOAs. Each element of $\boldsymbol{\tau}$ is associated with its respective element of the vector $\boldsymbol{\kappa}_{i_p}$, which indicates the reliability of each TDOA estimation. By sorting the values of $\boldsymbol{\kappa}_{i_p}$ from the highest to the lowest, we obtain a blind reliability ranking for the estimated TDOAs. Figure 3 shows the blind TDOA estimation obtained for each channel $j$ together with the blind reliability ranking shown in colour scale.

A non-blind comparison is also performed in Fig. 3 by showing the distance between the known acoustic source and the $j^{th}$ DAS acoustic channel. A qualitative comparison between distance and

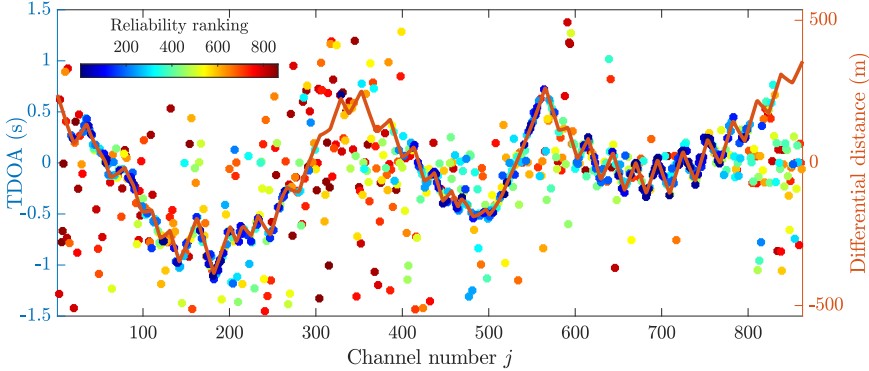

**Fig. 3 Blind TDOA estimation.** The blindly estimated time difference of arrivals (TDOAs) using the selected pilot trace and all the DAS data channels are compared with the non-blind differential distances calculated from the actual positions of the acoustic source and sensing fibre. The distance to the selected pilot trace channel is subtracted to visually compare the relative distances and TDOAs with respect to the pilot channel. Since the wave propagation velocity $v$ is unknown, TDOAs and distances cannot be compared with the same time or distance unit. The colour scale represents the blind reliability ranking. This ranking is composed so that the most reliable TDOA is ranked in the 1st place and the least reliable one in the 863rd position.

TDOAs can be performed if we assume a constant propagation velocity $v$. Results point out that channels with better reliability ranking (dark blue dots) present a very good blind TDOA estimation, closely following the shape of the relative distance curve with high precision. Channels with worse reliability ranking are however scattered around the expected value (dots changing colour towards dark red), indicating a higher estimation uncertainty. This confirms that the reliability ranking is a good method to blindly rank TDOA estimations. Note that some of the lower-quality channels are all located in neighbouring areas (see the position of channels 280–400 and 800–863 in the upper-right corner of Fig. 1a) and they are presumably affected by the mechanical properties of the soil in these areas, which may have led to poor strain coupling to the optical fibre. Since unreliable DAS channels must be discarded, the actual number of channels to be used by a beamformer must be blindly identified, as described hereafter.

**Acoustic signal enhancement through spatial filtering.** Iterative delay-and-sum beamforming[23,41] (see "Methods") is here used to implement a spatial filter that enhances the measured acoustic waveform quality, when compared to the most reliable DAS channel measurement. Given the poor coherence between some acoustic channels, due to distortions and different sensitivities, only a fraction of DAS channels must be used, resulting in a sparse sensor array for beamforming. Adding channels having distorted waveforms due to echoes and reverberations might impair the acoustic signal quality and therefore only traces with low distortion must be used by the beamformer. For this, a strategy must be defined to blindly find which DAS channels are useful for beamforming-based signal enhancement. As described in Fig. 4, the proposed method addresses this problem by generating different beamforming signals based on diverse numbers of input acoustic signals, followed by a blind evaluation of the $\beta$ indicator. The signal with the best $\beta$ indicator is then selected as the beamformer output. Since DAS channels are ordered according to their blind similarity indicator $\kappa_{i_p}$, each group of $\triangle m$ DAS channels added to the process (being $\triangle m = 20$ in this case) is less reliable than the previous one. Although adding new channels to the beamformer normally results in a noise reduction, the signal enhancement turns out to be limited by potential distortions introduced when adding less reliable channels.

By using the same process to evaluate the best pilot trace, the RMS reliability indicator $\beta_m$ of each $BF_m$ signal can be blindly estimated. Figure 5a shows the obtained $\beta_m$ indicator as a function of the number of channels $m$ used by the sparse beamformer. Results point out that the maximum indicator occurs with $m^* = 41$ channels, decaying after adding more channels. These 41 most reliable DAS channels are nonuniformly distributed in space, as depicted in Supplementary Fig. 11a. This approach allows us to blindly estimate the number of channels at which the indicator $\beta$ begins to deteriorate. Figure 5a also shows the SNR of each beamforming signal $BF_m$. This SNR is calculated in the frequency domain, where the signal noise is estimated by integrating the spectrum of a temporal window along the first 5 s, when there is no acoustic wave. Note that the behaviour observed for the SNR might wrongly lead to the conclusion that using all acoustic channels would result in a better signal enhancement. This is however incorrect since the SNR increases only due to the large background noise reduction, thanks to conventional trace averaging, whilst the signal might be highly distorted due to the combination of poor (incoherent) and good (somehow coherent) quality channels. This way, the SNR improvement is only meaningful if the resulting signal maintains or improves its similarity with respect to the reference. Figure 5b presents a non-blind validation of the selection method based on the calculation of a normalised $\kappa_{rm}$ indicator. Thus, it is possible to determine the similarity between each $BF_m$ and the reference acoustic signal r, helping us to assess the signal quality obtained with the blind proposed method. Results indicate that the beamforming signal blindly selected does not necessarily correspond to the one with the highest similarity, but it is among one of the best. Results also confirm a relative improvement of about 15% in the similitude of the most reliable DAS waveform (i.e., pilot trace) with respect to the emitted acoustic signal by reducing the impact of reflections and reverberations. Note that reaching a similarity of 100% would lead to a perfect representation of the reference acoustic wave; however, reaching this level is practically impossible due to the dispersion and ground attenuation of high-frequency mechanical wave components[42]. The figure also demonstrates that when the number of channels used by the beamformer increases too much, adding channels with poor-quality measurements (i.e., less coherent), the resulting beamforming signal losses its similarity with the emitted acoustic signal. Supplementary Fig. 5 verifies the benefits of using delay-and-sum spatial filtering compared to simple acoustic trace averaging using no alignment, demonstrating a 2.7-fold improvement in the similitude obtained after beamforming. Figure 5b also shows the SNR for each $BF_m$ signal, verifying that the chosen number of channels used for beamforming leads to an SNR improvement of 5.5 dB with respect to the pilot trace.

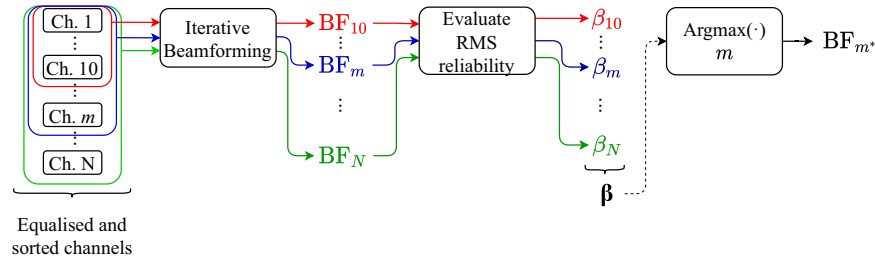

**Fig. 4 Signal enhancement procedure based on beamforming spatial filtering.** The first step of the process requires an amplitude equalisation (see "Methods") of the DAS channels to eliminate the effect of geometric spreading and propagation losses, so that they result with similar amplitude levels. After this, the DAS channels (ch.) are sorted based on their reliability indicator $\kappa_{i_p}$. Then, the beamforming signal $BF_m$ is formed using the $m$ channels with the best $\kappa_{i_p}$. In this case $m \in [1, 861]$, being selected with increments $\triangle m$ of 20 channels. This way delay-and-sum beamforming is applied to the most reliable channels, generating beamforming signals $BF_1$, $BF_{21}$, $BF_{41}$, and so on. The RMS reliability indicator $\beta_m$ is blindly calculated for each $BF_m$ signal. The beamforming signal with the best $\beta$ indicator is selected. The proposed method allows us to assess the impact of adding new $\triangle m$ channels in the beamforming signal because the quality of a $BF_{m+\triangle m}$ signal can be readily compared to a $BF_m$ signal, since the first $m$ DAS channels used in the processing are the same.

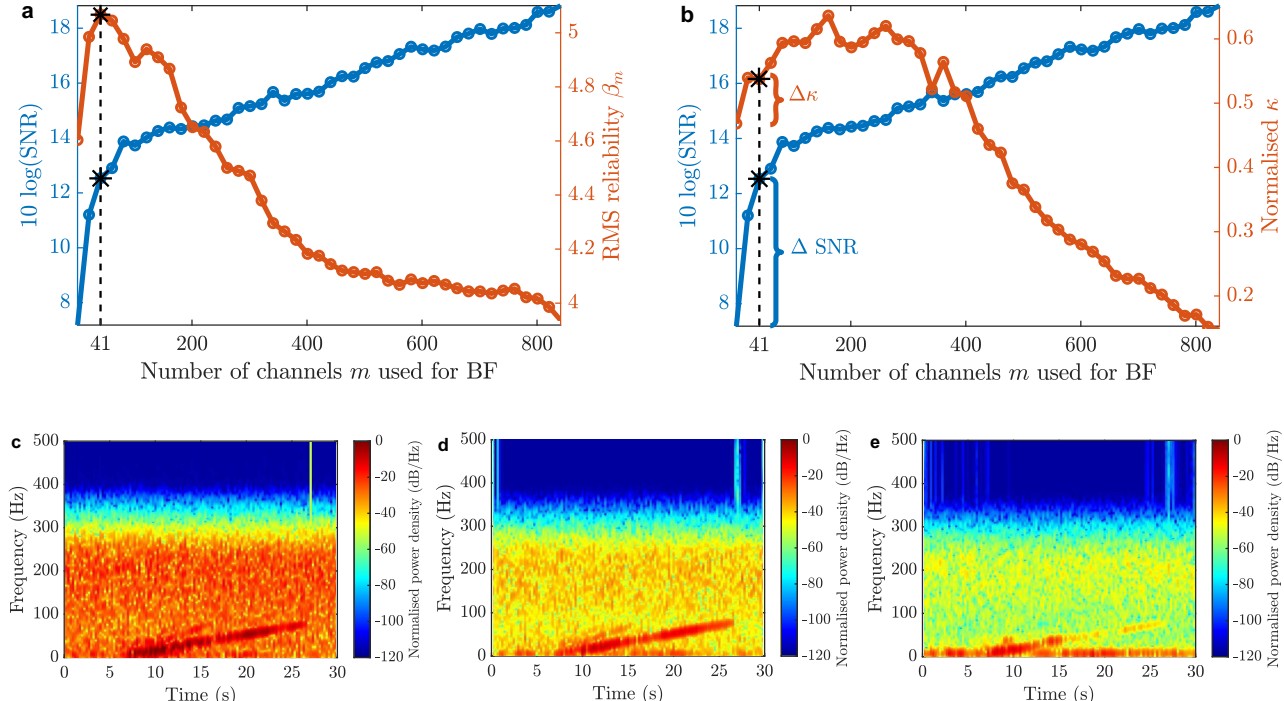

**Fig. 5 Blind and non-blind evaluation of the signal enhancement method. a** Blind RMS reliability indicator and SNR versus the number of channels used in the beamforming spatial filtering process. **b** Non-blind evaluation of the normalised indicator $\kappa_{rm}$ and SNR versus the number of channels used in the beamforming spatial filtering process. The normalised indicator $\kappa_{rm}$ is calculated between the reference acoustic signal r and each beamforming signal $BF_m$, normalised by the indicator $\kappa_{rr}$ based on the autocorrelation of the reference acoustic waveform. Spectrogram of beamforming signals generated for **c** $m = 1$, **d** $m^* = 41$, and **e** $m = 863$ channels.

Figure 5 also shows the spectrograms of the beamforming signals obtained for three values of $m$, where $m = 1$ corresponds to the pilot trace (Fig. 5c). Results reveal that for $m^* = 41$ (Fig. 5d), the measured chirped acoustic signal is reinforced, while the background noise is reduced. As the number of channels increases up to $m = 863$, the noise is greatly reduced at the cost of two undesirable effects (see Fig. 5e): (i) The use of several low-quality (incoherent) channels worsens the resulting beamforming signal, reducing the useful signal contrast with respect to the noise level, and (ii) the low-frequency interference that is present in some DAS channels is reinforced, as depicted in Supplementary Fig. 2.

**Acoustic source location estimation.** A modified hyperbolic triangulation[36] approach (see "Methods") is used to blindly find

the actual spatial coordinates of the acoustic source based on the estimated TDOAs. Similar to signal enhancement, the different acoustic sensitivities and distortions affecting each DAS channel along the sensing fibre[15–19] make the acoustic source location estimation a non-trivial problem. Using channels with unreliable TDOA estimations can significantly worsen the source location estimations. Figure 6 illustrates the process here proposed to identify the most often source location estimation using the most reliable channels in a fully blind and general manner, including the case when the acoustic velocity in the propagation media is unknown (assuming the speed is constant and the same for all DAS channels).

Based on the procedure described in Fig. 6, the spatial coordinates of the acoustic source and propagation velocity $\mathbf{x}^* =$

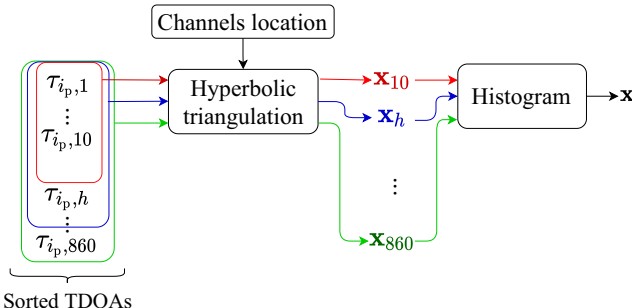

**Fig. 6 Blind source location strategy.** First, the TDOAs are ordered according to their reliability indicator $\kappa_{i_p}$, and a hyperbolic triangulation process is applied. We define the vector $\mathbf{x}_h = (\hat{x}_h, \hat{y}_h, \hat{v}_h)$ to contain the two-dimensional estimation of the source location $(\hat{x}_h, \hat{y}_h)$ and the estimation of the average propagation velocity $\hat{v}_h$, using $h$ TDOA estimations, with $h \in [5, 860]$ and increments of $\triangle h = 5$ channels. This way, for instance $\mathbf{x}_5$ corresponds to the estimation obtained with the 5 most reliable channels, $\mathbf{x}_{10}$ uses the same initial 5 channels adding the next 5 most reliable ones, and so on. With this procedure, we can analyse the effect of adding $\triangle h$ channels and blindly detect when the estimation starts to worsen. Although the number of channels to be considered in the processing is expected to be much lower than 860, this large number of DAS channels is here used for illustrative purposes. In the process, we define $\mathbf{X} = [\mathbf{x}_5, \mathbf{x}_h, \dots, \mathbf{x}_{860}]$ as a vector that includes all the estimations of the acoustic source position and propagation velocity obtained for different number of channels $h$. Once the vector $\mathbf{X}$ is obtained, the histogram of $\mathbf{X}$ is calculated for each component, i.e., for $\hat{x}_h$, $\hat{y}_h$ and $\hat{v}_h$, defining the histograms $H_x$, $H_y$ and $H_v$, respectively. By using several values of $h$ reliable channels, similar location estimations $\mathbf{x}_h$ can be obtained until more unreliable channels are included in the processing. This way, using a large enough number of reliable channels, the most common estimations $\mathbf{x}_h$ will be highly reliable, while adding unreliable channels will lead to different estimations. Therefore, we find a trustworthy source location by choosing the most repeated estimation.

$(x^*, y^*, v^*)$ are estimated by choosing the mode of the respective histogram $H_x$, $H_y$ and $H_v$ (i.e., the most repeated $x$, $y$ and $v$ estimations). Figure 7a–c shows the estimated values of $\hat{x}_h$, $\hat{y}_h$ and $\hat{v}_h$, respectively, as a function of the used number of channels $h$. Results reveal that in all three cases, the estimated values remain relatively constant in the range of approximately $30 < h < 400$. The use of less than 30 channels seems to be insufficient to achieve a reliable estimation, while the use of more than 400 channels affects the estimation due to the inclusion of low-reliability channels. Figure 7d–f shows the respective histograms of the three estimated variables. Note that, for the sake of illustrative purposes, these histograms include all the performed estimation, i.e., including the participation of up to 860 (i.e., almost all) DAS channels.

To validate the blind process, Fig. 7 also includes the real acoustic source coordinates, as red dashed lines. We can observe that the peaks of the histograms $H_x$ and $H_y$ (at $x^* = 243.5$ m and $y^* = 952.5$ m, respectively) are closely located to the actual acoustic source position $(x_s, y_s) = (240.6 \text{ m}, 953.6 \text{ m})$, validating the good precision of the method. This verification cannot be easily performed for the estimated velocity ($v^* = 342.5 \text{ m s}^{-1}$) since there is no single reference value for this parameter; however, the estimated value is within the range of velocities reported in a prior study of the here analysed Brady Hot Springs area[42].

The performance of the blind near-field method is assessed by quantifying the absolute error $e_{\text{hist,s}}$ between the estimated location $(x^*, y^*)$ and the known position of the acoustic source $\mathbf{x}_s = (x_s, y_s)$, according to

$$e_{\text{hist,s}} = \sqrt{(x^* - x_s)^2 + (y^* - y_s)^2} \tag{1}$$

To verify the impact of including different numbers of channels in the processing, the absolute error $e_{h,s}$ of the estimated location $(\hat{x}_h, \hat{y}_h)$ obtained with the $h$ most reliable channels (see Supplementary Fig. 6) is also calculated by an expression similar to Eq. (1) and shown in Fig. 8a. This figure also shows the actual estimation error $e_{\text{hist,s}}$ obtained from histograms (red dashed line), indicating that the estimation error stabilises around the error obtained by the histogram when using 30–400 channels, being consistent with results in Fig. 7. This result validates the proposed blind procedure based on the histograms, allowing us to identify the most recurrent estimation, which remains among the lowest obtained errors. Note that the achieved estimation error is 3.06 m, corresponding to a very small relative error of 0.83%, when compared to the distance $d_{\text{RMS,s}} = 367$ m between the acoustic source and the sensing fibre. Figure 8b shows the acoustic source position $\mathbf{x}_s$ (red square) and the estimated location $\mathbf{x}^*$ (black circle), verifying that the proposed blind method identifies with high precision the source spatial coordinates in the $(x, y)$ plane. As an example, Supplementary Fig. 11b, c shows the best 100 and 200 channels used for estimating the source location. Note that channels are randomly distributed along the sensor, highlighting the nonuniform spacing between DAS channels used in different processing steps. This emphasises the need for a method to rank the channel quality such as the one proposed here, before applying the source location method.

**Generalisation and statistical analysis.** We must consider that the results in acoustic signal enhancement and source location highly depend on the features of the acoustic wave propagation, which could differ depending on the acoustic source position with respect to the sensing optical fibre[15,16,19]. For this reason, the results obtained by the blind beamforming-based processing are statistically analysed using 50 different acoustic source positions (see Fig. 1a) and the non-blind verification procedure described before.

Figure 9a shows the statistical distribution of the ranking position associated to the pilot traces blindly selected for all datasets. Results reveal that 50% of pilot traces are ranked within the 9 channels with the highest similarity with respect to the acoustic reference signal, out of the 863 channels, while 25% of pilot traces are statistically ranked below the position 2.25. The figure also verifies that when the best SNR measurements are used as pilot traces, these are ranked in much worse positions, demonstrating that high SNR measurements are not necessarily similar to the reference acoustic signal. It is worth noticing that all pilot traces obtained using the blind reliability indicator are ranked below the 50$^{\text{th}}$ position, while only 25% of pilot traces selected based on their SNR show comparable levels of similarity. The better results obtained by the proposed method can be justified by the use of reliability indicators based on similarity and phase correlations rather than on signal amplitudes.

Additionally, the spatial filtering performance is assessed by evaluating the simultaneous enhancement of both SNR and similarity indicator $\kappa$ compared to the best channels (i.e., the respective pilot traces). Figure 9b shows the improvement obtained in both SNR and similarity indicator, where each point represents the result obtained for each of the 50 analysed datasets. Boxplots of each of these variables are also shown. The red dotted lines indicate the zero-value point on each axis, thus separating

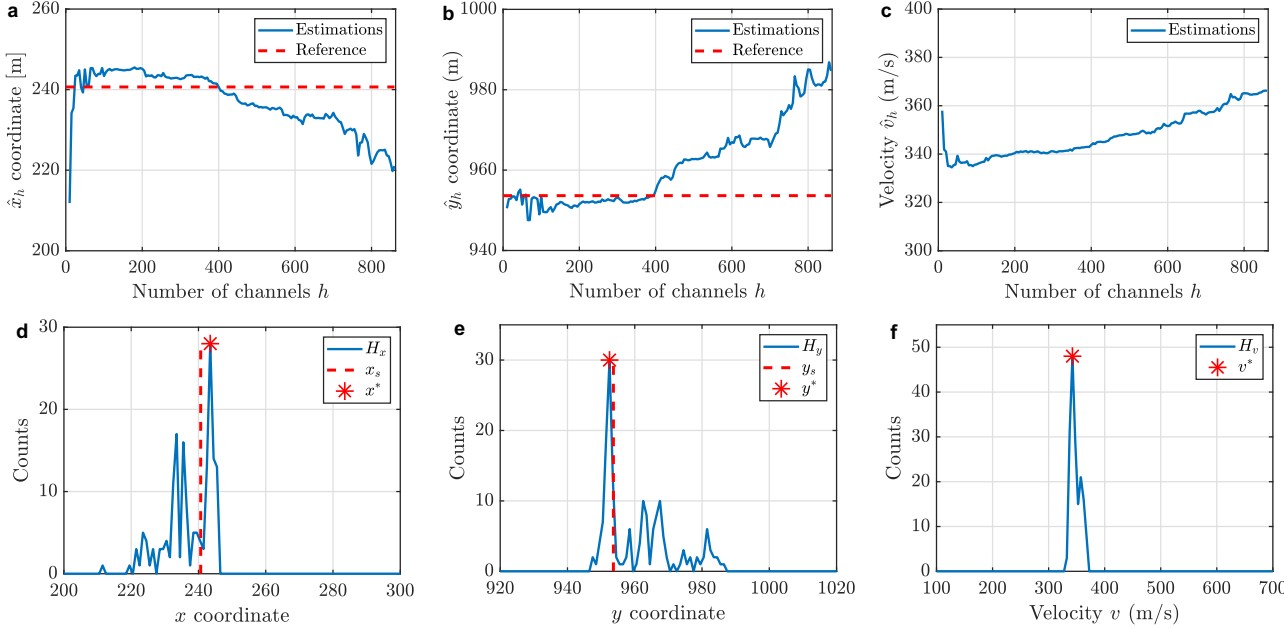

**Fig. 7 Source location estimation using different number of sensors _h_.** Estimated **a** $\hat{x}_h$ coordinate, **b** $\hat{y}_h$ coordinate, and **c** average propagation velocity $\hat{v}_h$, as a function of the number of channels $h$ used in the process. The histogram of each estimated parameter uses a bin width of 1 m for the histogram **d** H$_x$ and **e** H$_y$, and a bin of 5 m s$^{-1}$ for the histogram **f** H$_v$. Dashed red lines correspond to the actual source coordinates $x_s$ and $y_s$. The red stars indicate the blindly selected coordinates and velocity based on the mode of the histograms.

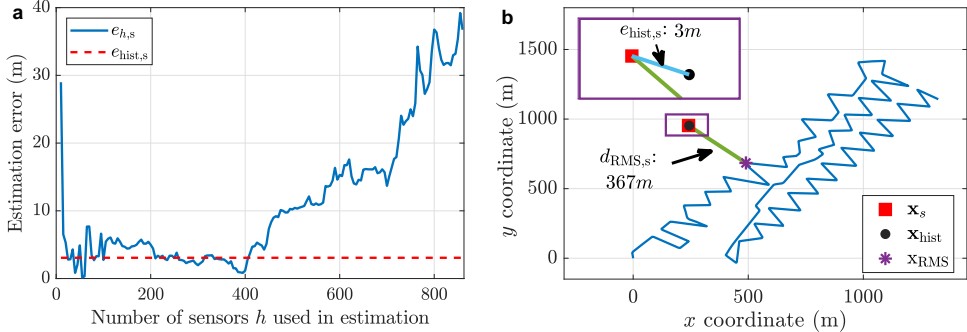

**Fig. 8 Error in the blind source position estimation. a** Non-blind error estimation based on Eq. (1) as a function of the number of sensors $h$ used in the process. Supplementary Fig. 6 shows location estimation for different channels $h$, comparing the absolute error with respect to the sensing optical fibre and actual source position. **b** Illustration of the estimated **x***(black circle) and actual **x**$_s$ (red square) source location, compared to the sensing optical fibre location (using the DAS channel with the highest RMS value **x**$_{RMS}$ as a reference point). The Inset shows a zoom-in of the relevant area of interest, showing a very small error of 3 m in the estimation compared to the 367 m separating the acoustic source from the sensing fibre.

areas with impairment and improvement. More than 75% of the datasets show improvements of both indicators (red circles), implying that their SNR and similarity with the reference signal increase simultaneously. Note that more than 50% of the datasets have an SNR improvement between 4.36 dB and 18.54 dB while maintaining or increasing the similarity with the reference signal. In addition, based on the similarity improvement shown in Supplementary Fig. 10, more than 50% of the cases show a relevant similarity improvement over 18.05%.

Finally, a non-blind statistical analysis of the error in estimating the acoustic source position is also carried out by comparing the estimations with the actual source coordinates. Figure 9c shows a boxplot with the statistical results of the relative errors obtained for all datasets. Note that, using the proposed blind estimation method, 75% of the analysed datasets have a relative error below 23.8%, while 25% of the datasets resulted with relative errors below 4.1%. Supplementary Fig. 12 shows a source location error map for all analysed datasets. The original source

positions are shown together with the position estimated by the proposed algorithm. Note that for most of the datasets, the estimated position is very close to the reference position. The figure allows us to dimension that even large estimation errors (e.g., over 200 m) are not so far away from the real position of the acoustic source given the dimensions of the survey area, demonstrating the high precision of the proposed method.

## Discussion

Note that the approach here demonstrated as a proof of concept is based on one of the simplest array signal processing techniques[22–24,41] applied to DAS recordings, and therefore there is margin for further improvements, especially for complex acoustic wave propagation scenarios, like the present one. Indeed, simpler propagation conditions would naturally lead to better processing performance compared to the one reported here, due to the eventual higher coherence among DAS channels. In addition, the methods for signal enhancement and source

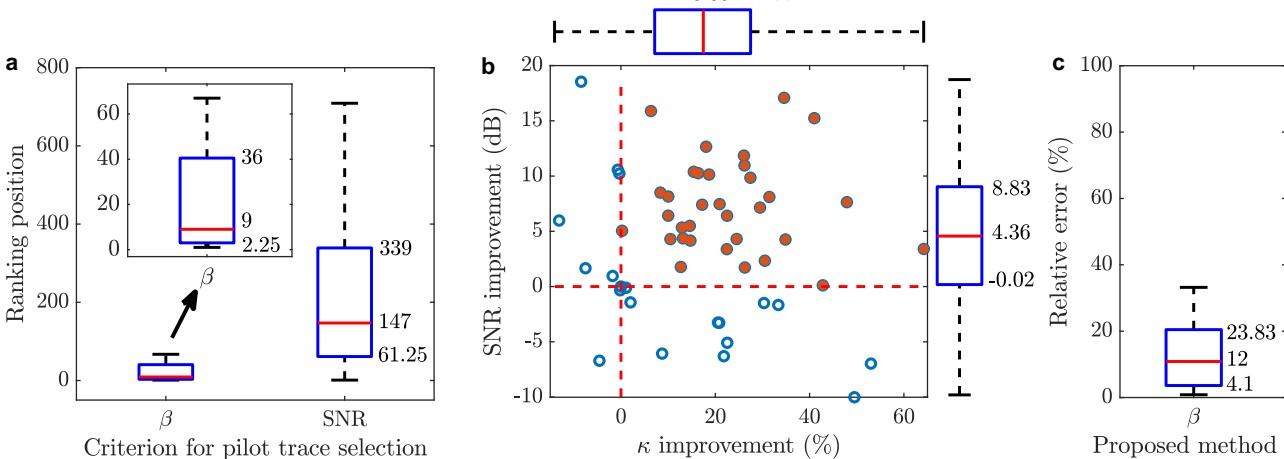

**Fig. 9 Statistical results. a** Ranking of the pilot traces selected over 50 datasets with different source locations using the proposed blind RMS reliability indicator $\beta$ and the channel with the best SNR. The boxplots in the figure indicate the 25th, 50th and 75th percentile of the ranking position distribution. For the sake of visualisation, the inset shows a zoom-in of the results based on the blind RMS reliability indicator $\beta$. **b** SNR enhancement and improvement of the similarity indicator $\kappa$ for the 50 analysed cases. **c** Relative error of the proposed blind source position estimation.

position estimation can be iteratively implemented to improve TDOA estimations and potentially achieve better results. Supplementary Fig. 7 shows that, despite the existing errors in the source location estimation, the TDOAs calculated using the estimated source position are quite close to those non-blindly calculated based on the actual source location, indicating that potential further improvements could be achieved by an iterative process.

It is worth mentioning that in the present proof of concept, sparse beamforming spatial filtering has been exploited to improve the waveform representation emitted by a single point seismic acoustic source, reducing distortions caused by reflections and reverberations arriving to the sensing fibre with different angles. However, the spatial filtering capabilities of the method can be further explored in many applications to clearly discriminate the acoustic waveforms emitted simultaneously by several acoustic sources, which combined with hyperbolic triangulation can allow us to identify their actual 2D or even 3D coordinates with no specially designed optical fibre installation geometries, provided there exist good angular diversity of the optical fibre orientation. Indeed, it must be noted that in linear (straight) fibre installations, some ambiguities will arise due to the geometric symmetry of the fibre, impeding the array signal processing to identify the side of the optical fibre where the acoustic source is located. However, the use of straight linear fibres may also be inadequate for real-field DAS applications, due to the directional response of a DAS sensor. In particular, if the acoustic signal is broadside to the cable, no strain will be measured. This is an issue affecting all DAS sensors and not only for the method here proposed. Therefore, robust DAS measurements might require the use of an optical fibre installation with different orientations, making the DAS monitoring robust against random directions of the acoustic wave arrival (although some local measurements will still have null or poor response). Incidentally, the use of different fibre orientations also benefits the method here proposed to better identify the actual source position without triangulation ambiguity. Nevertheless, note that the performance of the method highly depends on the acoustic properties of the propagation medium and the positioning of the acoustic sources and sensing optical fibre, which define the acoustic attenuation and reflections, among other phenomena occurring during propagation. In addition, instead of the blind TDOA estimation approach here applied to localise the position of the

source, TDOAs can be exhaustively scanned using simple geometry on an entire 2D or 3D region, allowing for the mapping of the total acoustic field existing in a large area or volume and emitted by several sources. This way, the spatial filtering capabilities here demonstrated based on a sparse DAS array configuration can be potentially exploited to implement, for instance, acoustic cameras based on DAS technology, with an optical fibre that does not need to be installed over the analysed zone.

Regarding execution time of this first demonstration, the most demanding part is the blind pilot trace selection since phase cross-correlations among all channels are calculated. Using an Intel(R) Core(TM) i7-9750H CPU @ 2.60 GHz, this calculation takes about 5.6 min in Matlab, followed by 3.5 min required for spatial filtering. All other processing times, including the source position estimation, remained below 1 s. Note however that cross-correlations can be very efficiently computed using Fast Fourier transforms in graphical processing units (GPUs) using parallel processing based on compute unified device architecture (CUDA)[39], potentially providing a significant reduction of the overall computation time. Determining how many times the current calculation needs to be accelerated for real-time processing highly depends on the target response time required for a specific application. For instance, if the execution time is reduced to equal the acoustic signal length (30 s in this case), an acceleration factor of 18 would be required. On the other hand, improving the current processing time in a factor 500, for example, would lead to an execution time of about 1 s. Note that these levels of acceleration, compared to our CPU-based Matlab implementation, can be easily achieved with a proper CUDA programming using GPUs[43]. It is also worth pointing out that here all DAS channels are cross correlated and included in the processing for illustrative purposes and to verify the detrimental impact of adding channels with low-quality (low coherence) measurements. Therefore, the mentioned execution times are only for the worst-case scenario when all DAS channels are considered in the array signal processing. In practice, the computational time can be reduced by preselecting and discarding those channels that are clearly not reliable, as well as by setting a stopping criterion based on the identification of the optimal number of channels before reliability indicators reduce. In addition, it should be noted that, in this case, the cross-correlation window to evaluate channels is 30 s, corresponding to 30k data points. In a general context, this window size could be still further

optimised depending on the length and spectral content of the signals to be processed. This would secure that the window captures the main features of the measured signal to provide reliable estimations of the TDOAs and channel quality.

It is important to note that in this proof of concept the beam characteristics of the spatial filter are not designed nor steered to a predefined position, and therefore no spatial information about the sensing fibre is required in this approach. Here only the estimated TDOAs are used to enhance the best measured acoustic signal, represented by the pilot trace. Considering the large number of sophisticated array processing techniques existing in the literature[22–24,44,45], we are confident that novel approaches will soon emerge to improve the directivity and performance of near- and far-field beamforming techniques for DAS applications. For instance, making use of the optical fibre positioning and local fibre orientations, complex weights can be designed for advanced beamformers to control the directivity pattern and increase the selectivity of the spatial filtering results. Note however that the uneven response and low coherence of DAS acoustic channels would affect any standard beamforming method, which normally assumes identical sensor responses; and therefore, the here demonstrated blind ranking strategy based on the reliability of DAS channels can be used to adapt standard beamforming techniques for DAS applications. In addition, further improvements could be obtained in some scenarios by pre-processing DAS measurements using deconvolution and dereverberation techniques to remove acoustic reflections[19–21]. We believe that the proposed technique is a starting point opening new class of acoustic processing strategies to enhance the capabilities of distributed acoustic sensors, but also other technologies like arrays of fibre Bragg gratings[46,47] and multiplexed interferometric fibre sensors[48,49], to measure the acoustic field existing outside the optical fibre.

## Methods

**Evaluation of channel reliability**. Due to the nonuniform sensitivity of DAS acoustic channels and the detrimental impact of reflections and reverberations, some channels result in distorted measurements[15–19]. Although this is a fixed feature of some channels (e.g., due to poor coupling between the ground and optical fibre or Rayleigh intensity fading), low-quality measurements could randomly occur along the sensing fibre depending on the acoustic wave position and the presence of reflections and reverberant effects during wave propagation. High-quality channels (i.e., with low distortions) are here considered more reliable to estimate TDOAs. Searching the most reliable channels may be trivial if we know the emitted acoustic signal, since this can be directly compared with the waveform under evaluation. However, in a general DAS scenario, this search must be performed in a completely blind manner following the strategy described in Fig. 10a. This is here performed by estimating the level of mutual similarity between the input channel $i$ and all other channels $j$, with $j = 1, \dots, N$, being $N$ the total number of channels by calculating the phase cross-correlation function $\text{PCCF}_{ij}$. Note that the use of phase cross-correlation is much more suitable than amplitude

cross-correlation to assess the similarity and phase coherence between two broadband waveforms[38,39].

For each $\text{PCCF}_{ij}$, two parameters are obtained: (i) the lag $\tau_{ij}$ of the main correlation peak (highest absolute amplitude), corresponding to the relative time delay between waveforms, and (ii) the ratio between the maximum absolute value and the RMS value of a window $W$ surrounding the main correlation peak. This ratio is here defined as an indicator $\kappa_{ij}$ and corresponds to the peak-to-root-mean-square ratio (PRMSR)[50], defined as:

$$\kappa_{ij} = \frac{\max(|\text{PCCF}_{ij}|)}{\text{RMS}(W)} \qquad (2)$$

with $\text{RMS}(W) = \sqrt{\frac{1}{2L}\sum \text{PCCF}_{ij}^2(a)}$, where the sum is performed between $a \in [\tau_{ij} - L, \tau_{ij} - \delta t] \cup [\tau_{ij} + \delta t, \tau_{ij} + L]$ (with $\delta t$ being the sampling period), i.e., through a window of length $2L$ centred in $\tau_{ij}$ and discarding the main correlation peak value at $\tau_{ij}$, as described in Fig. 10b. Note that the PRMSR is similar to the SNR concept, but applied to the PCCF[50].

A high value of $\kappa_{ij}$ represents a high-amplitude $\text{PCCF}_{ij}$ peak with low correlation sidelobes, indicating a high level of certainty in the relative delay estimation $\tau_{ij}$ between channels $i$ and $j$ (see Supplementary Fig. 3a). A low value of $\kappa_{ij}$ indicates that the PCCF does not have a dominant main correlation peak, which could occur due to the following three reasons:

- There is not good correlation between both channels, so there is no peak that stands out in the $\text{PCCF}_{ij}$ (see Supplementary Fig. 3b).
- There is good correlation, but there are other peaks of considerable amplitude within the window $W$ in the $\text{PCCF}_{ij}$, which increase the RMS value of the window. This might happen in highly reverberant scenarios and multipath propagation conditions due to multiple reflections (see Supplementary Fig. 3c).
- Both channels contain narrowband signals within the same frequency range, so their $\text{PCCF}_{ij}$ has a periodic behaviour without a single main correlation peak that stands out with respect the surrounding area (see Supplementary Fig. 3d).

This way, the indicator $\kappa_{ij}$ allows us to find channels with predominant and isolated correlation peaks, representing a pair of broadband signals similar to each other with low reverberation and distortions.

In the TDOA estimations, the similarity indicator $\kappa_{ij}$ is calculated between a given pilot trace candidate $i$ and all other channels $j$, resulting in a $1 x N$ vector defined as $\boldsymbol{\kappa}_i = [\kappa_{i,1}, \kappa_{i,2}, \dots, \kappa_{i,N}]$, which contains all the individual reliability indicators for the candidate pilot trace channel $i$. Obtaining the RMS value of the vector $\boldsymbol{\kappa}_i$, the information is summarised in a single indicator defined as $\beta_i$, which represents the RMS reliability of the TDOA estimations of all DAS channels with respect to the candidate channel $i$. The $\beta_i$ indicator is obtained as:

$$\beta_i = \sqrt{\frac{1}{N}\sum_{j=1}^{N}\kappa_{ij}^2} \qquad (3)$$

where $j \neq i$, to discard the value $\kappa_{ii}$ corresponding to the PRMSR of the autocorrelation of the channel $i$. The channel with the highest $\beta_i$ indicator is selected as the most reliable one.

It is important to note that the reliability indicator $\kappa$ is calculated between two channels, whereas the indicator $\beta$ evaluates the reliability comparing one channel with all other channels or waveforms. Both are used to measure the reliability with respect to a reference signal or channel, but $\kappa$ corresponds to the individual reliability between channels and $\beta$ to the RMS reliability between a reference signal and all other channels. In this way, when performing non-blind validations in Fig. 5b, the value of $\kappa_{rj}$ is calculated to assess the individual reliability of each

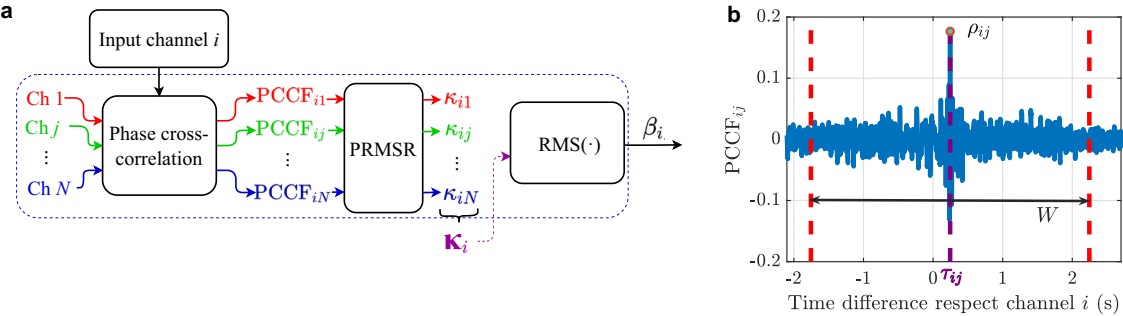

**Fig. 10 Evaluating the reliability of an input channel. a** Procedure to blindly evaluate the RMS reliability of a given channel $i$, based on the calculation of the phase cross-correlation with respect all other DAS channels $j$. The individual reliability indicator $\kappa_{ij}$ is calculated as the peak-to-mean square ratio (PRMSR) of the phase cross-correlation function $\text{PCCF}_{ij}$. By obtaining $\kappa$ for a given channel $i$ with respect to all other $j$ channels, the vector $\boldsymbol{\kappa}_i$ is generated. Then, the indicator $\beta_i$ is obtained as the RMS value of the vector $\boldsymbol{\kappa}_i$. **b** Example PCCF, indicating the relative time delay $\tau_{ij}$, correlation peak amplitude $\rho_{ij}$ and window $W$.

channel $j$ with respect to the reference signal r, corresponding to the known emitted acoustic signal. In that case, the values obtained are normalised with respect to the PRMSR value ($\kappa_{rr}$) of the autocorrelation of the signal r.

**Amplitude equalisation**. To eliminate the differences in the amplitude response of DAS channels, as shown in Supplementary Fig. 1, a multiplicative constant $C_j$ must be applied to each channel for equalisation. The constant $C_j$ can be obtained by minimising the mean square error (MSE) of the spectrum amplitude among channels through least squares. Using the first channel as a reference, $C_j$ is obtained by minimising the following expression:

$$\text{MSE} = \sum_{\omega} \left\{ A_1(\omega) - C_j A_j(\omega) \right\}^2 \quad (4)$$

with $j = 2, \dots, N$, where $N$ is the total number of DAS channels and $A_j(\omega)$ is the spectrum amplitude of the channel $j$. The solution $C_j$ that minimises the error is given by[51]:

$$C_j = \frac{\sum_{\omega} A_1(\omega) A_j(\omega)}{\sum_{\omega} A_j^2(\omega)} \quad (5)$$

Once the multiplicative constant is obtained for each channel, the amplitude spectra is equalised as $A'_j(\omega) = A_j(\omega) C_j$.

**Near-field delay-and-sum beamforming**. Near-field beamforming considers that the acoustic source is located at a distance $d < 2L^2/\lambda$ from the sensor array[22–24] (considering a given sensor as a reference), where $L$ is the largest dimension of the sensor array (in this case, the largest dimension of the region covered by the sensing optical fibre) and $\lambda$ is the acoustic wavelength. Under these conditions the wavefront is assumed to be spherical, instead of a plane wave as in far-field approaches. Therefore, the acoustic signal arriving at each DAS channel has different amplitudes and angles of arrival. Thus, each DAS channel measures a signal $y_m(t)$ given by:

$$y_m(t) = f(d_m) s(t - t_m) \quad (6)$$

where $s(t)$ is the emitted acoustic signal, $f(d_m)$ is a function describing the amplitude attenuation as a function of the distance $d_m$ between the source and each sensor $m$, and $t_m = d_m/v$ is the time delay between the source and sensor $m$ resulting from the propagation of the acoustic signal at a velocity $v$.

Delay-and-sum[23,41] is one of the best known and simplest beamforming techniques and is used here as a proof of concept of the proposed method. The beamforming signal BF($t$) generated by the delay-and-sum method is described as[23,41]:

$$\text{BF}(t) = \frac{1}{M} \sum_{m=0}^{M-1} w_m y_m(t + \tau_m) = \frac{1}{M} \sum_{m=0}^{M-1} w_m f(d_m) s(t - t_m + \tau_m) \quad (7)$$

where $M$ is the number of reliable DAS channels used in the processing, $\tau_m$ corresponds to a time delay applied to synchronise the signals, and $w_m$ is a weight added to each channel.

From Eq. (7) it is clear that by choosing $\tau_m = t_m$ and $w_m = f^{-1}(d_m)$ the emitted acoustic signal $s(t)$ can be perfectly recovered. Therefore, when using near-field delay-and-sum, we must first synchronise each DAS channel, then remove the effect of attenuation, and finally combine all DAS channels to obtain the beamforming output signal BF($t$) $\sim s(t)$. Note that in the approach of this work, the value of $\tau_m$ is blindly estimated by calculating phase cross-correlations, and $w_m$ corresponds to $\text{sign}\left(\max\left(\text{PCCF}(\tau_m)\right)\right) \cdot C_m$, where $C_m$ corresponds to the equalisation multiplicative constant for the $m^{\text{th}}$ channel, as defined in Eq. (5).

**Iterative delay-and-sum beamforming**. Based on the estimated TDOAs, we can use delay-and-sum beamforming[23,41] to align and combine DAS channels to generate an improved version of the pilot trace. TDOA estimations are actually affected by the different DAS channel sensitivities and distortions cause by reflections and reverberances. Therefore, delay-and-sum beamforming can be used to as a spatial filtering method to improve the quality of the pilot trace, which can then be used to obtain more precise TDOA estimations[40]. This procedure can be repeated iteratively using only a few reliable DAS channels to run the algorithm a given maximum number of times or to iterate until the variation in TDOAs estimation goes below a threshold, which can be defined, as for instance, based on the lag discretisation of the PCCF. In this case, we have set a maximum of 10 iterations. Supplementary Fig. 8 shows an example BF signal obtained through different iterations of the algorithm for a fixed number of channels, verifying that the RMS error of the estimated TDOAs tends and reaches zero before 10 iterations.

**Modified hyperbolic triangulation**. The hyperbolic triangulation method corresponds to one of the standard techniques to estimate the position of a source from the TDOAs between different sensors[36]. The method considers that the propagation velocity $v$ is known and uniform along the sensors. As for many DAS applications, this situation is not fulfilled by the DAS data used in this work, so the method is modified to include the velocity $v$ as one more variable to optimise. Considering the common uncertainties resulting in the TDOA estimations, the

method is performed by minimising the following cost function:

$$J_{\text{TDOA}}(x,y,v) = \sum_{i=2}^{h} \left| v\tau_{ri} - d_i(x,y) + d_r(x,y) \right| = \sum_{i=2}^{h} \left| v\tau_{ri} - \Delta d_{ri}(x,y) \right| \quad (8)$$

where $h$ is the number of estimated TDOAs being used, $v$ is the propagation velocity, $\tau_{ri}$ is the TDOA estimated between a reference sensor $r$ and the sensor $i$, $d_i(x,y)$ and $d_r(x,y)$ are the distances between the source and the respective $i$ and r DAS channels, and $\Delta d_{ri}(x,y) = d_r(x,y) - d_i(x,y)$. The channel 1 is here chosen as the reference channel. Compared to the traditional hyperbolic triangulation[36], our cost function $J_{\text{TDOA}}$ in Eq. (8) depends on three variables instead of only the two spatial coordinates.

The objective of the algorithm is to estimate the source spatial coordinates $(x^*, y^*)$ and velocity $v$ that minimise the cost function $J_{\text{TDOA}}$. It is important to note that $\tau_{ri}$ corresponds to the blind TDOA estimation, using no information about the acoustic source and optical fibre position, while $\Delta d_{ri}(x,y)$ is obtained by using a possible guessed source position $(x, y)$ and the optical fibre spatial distribution to compute the distances.

The cost function in Eq. (8) can be considered as a data fitting process, for which we want to estimate $(\hat{x}, \hat{y}, \hat{v})$ to make the parametric function $\Delta d_{ri}(x,y)/v$ to fit the noisy data. Note that outliers may appear in channels containing poor-quality measurements, due to distortions caused by reflections and reverberances. For this reason, we have also modified the standard hyperbolic triangulation method to use the L1 norm for the cost function, instead of the L2 norm employed by the standard approach. This is because the L2 norm squares each term, giving too much relevance to outliers, while the L1 norm results in outliers having lower impact.

A nonlinear optimisation algorithm known as trust region[52,53] is here used to minimise the cost function. The algorithm depends on an initial position to start the search. Supplementary Fig. 8 verifies that the actual position of the starting point has practically no effect on the performed estimation, this means that despite positioning the initial point at different spatial coordinates, the result in minimising the cost function is practically the same. Therefore, as a sake of simplicity, the starting location for the minimisation algorithm is here selected as the DAS channel having the highest RMS value, since this channel could eventually be closer to the acoustic source or presumably have direct line of sight.

## Data availability
The source data files containing the DAS measurements used in this work have been obtained from the online database "PoroTomo Natural Laboratory Horizontal and Vertical Distributed Acoustic Sensing Data" University of Wisconsin (2017). Retrieved from https://doi.org/10.15121/1778858.

## Code availability
The codes generated during the current study are available from the corresponding author on reasonable request.

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

## Acknowledgements

This work was supported by ANID Chilean National Agency for Research and Development, under Projects Fondecyt Regular 1200299 and Basal FB0008. The work of F.M. was also supported by the Dirección de Postgrado y Programas (through Convenio PIIC: 008/2020) of Universidad Técnica Federico Santa María.

## Author contributions

M.A.S. conceived the concept of blind near-field beamforming for DAS measurements. F.M. and M.A.S. developed the concept, established and designed the methodology. F.M. implemented all algorithms and defined all modifications with respect to standard approaches. F.M. and M.A.S. analysed the data and results. M.A.S. supervised the project. F.M. and M.A.S. wrote and revised the manuscript.

## Competing interests

The authors declare no competing interests.
