## [Peer Review File · Nature Communications]

REVIEWER COMMENTS

Reviewer #1 (Remarks to the Author):

In this contribution, the authors proposed a blind near-field array signal processing method based on fiber-optic distributed acoustic sensor (DAS). DAS has many sensing channels, but not all channels could provide valid and correct information. In view of this, this paper gives a method to evaluate and rank the reliability of each channel without prior knowledge, from which the pilot is selected. Then the pilot trace is improved by delay-and-sum beamforming further. The authors organized contrastive analysis to verify their proposed method with significant results.

The reviewer also have some questions:

1.The proposed method involves lots of calculation. How much time did the authors take for data processing?

2.The location of vibration source via DAS is not a new research field. The innovation of the manuscript mainly lies on the improved method for the selection of pilot trace, while the PCCF, time delay of arrival (TDOA), and beamforming methods have already been widely adopted.

3.The fine processing of the data usually means that the adaptability is reduced. Can the authors prove their method in other scenarios with superior performance?

4.In line 494, the author wonders if the Eq. (5) should be Eq. (7).

5.In Ku, Emery M., and Gregory L. Duckworth. "Tracking a human walker with a fiber optic distributed acoustic sensor." Proceedings of Meetings on Acoustics ICA2013. Vol. 19. No. 1. Acoustical Society of America, 2013, the beamform method is utilized in localizing footfalls. So the reviewer wonder whether the method is still useful for the shock signal which lasts a short time like footstep?

Reviewer #2 (Remarks to the Author):

Dear Marcelo and Felipe,

It has been a pleasure to read your paper submission, it was very well presented and the processing steps clearly described with interesting results.

The submission demonstrated a novel fully blind process to select high quality DAS channels to 1) improve the signal to noise ratio, and 2) determine accurately the signal source location through beamforming techniques. It was particularly interesting to see the statistics from the analysis of the data sets at the 50 different source locations.

There are a few points which I'd like to raise.

In figure 3 there are two notable areas 1) between channel 280 and 400 and 2) towards the end of the fiber, channel 800+, where the reliability ranking is poor i.e. very few dark blue dots. Do you have an explanation for why these areas don't show the high levels of reliability ranking as the majority of the installation? This might be the case where the signal is broadside to the cable and therefore the DAS response is reduced? It is difficult to tell without a better understanding of the installation. Can I suggest that figure 1(a) is labeled with channel 0 and ch862?

The test site had an 'unique' fiber installation pattern. As many of the installed fibers where this technique could be employed i.e. dark telecommunications fibers, are linear installations, what are your thoughts about performance in this, arguably more representative case. In the simplest case, it is worth mentioning that in a linear installation there will be ambiguity in the localization due to the symmetry i.e. which

side of the cable is the source.

Line 25 - I would suggest replacing the phrase 'kHz-to-MHz acoustic bandwidth' with 'wide acoustic bandwidth'. There are quite a few recent publications demonstrating MHz performance of DAS, so best to avoid putting general limits on the DAS acoustic bandwidth.

I suggest including the reference below, as far as I know this is the earliest DAS reference to signal localization through beamforming.

Ku, E. M. & Duckworth, G. L. "Tracking a human walker with a fiber optic distributed acoustic sensor." (2013). doi:10.1121/1.4800575.

Reviewer #3 (Remarks to the Author):

The manuscript presents a novel workflow to locate vibration sources using DAS data with imperfect array geometry and dataset. The work is innovative, the results are encouraging, and the discussion is comprehensive. I recommend the publication after minor revisions.

Several claims in the abstract can be confusing and misleading, especially considering most readers read it without knowing the context and method details. For example, Line 11 sounds like the workflow does not need to know the fiber location at all. On line 16, it is confusing to state no optical fiber is required (fiber still has to be close enough). I recommend the authors rephrase these statements.

It is a bit strange to put the "Method" section at the end. I had difficulties understanding the Result section without first reading the "Method" section. I recommend the authors reorder the sections.

The beamforming method that the authors proposed is based on the assumption that the source is isotropic, which radiates the same waveform in all directions. This may not be the case in the real world. For example, a shear source radiates waveforms with 180 phase shifts in different directions. It is better to make this assumption clear in the manuscript.

The workflow that the authors proposed is computationally expensive. I like that the authors admitted and discussed this in the Discussion section. However, I believe more information could be provided. For example, how long the cross-correlation window is, and how many times the current calculation needs to be accelerated (through GPU and parallel computing) in order to achieve real-time analysis, which is critical for this type of application.

Other minor comments:

Fig 8: it is good to highlight the channels used for the location estimation in figure b.

Fig 9: I recommend including a source location error map to better demonstrate the relation between estimation errors and relative positions between source and array.

Line 378: This is too optimistic. The accuracy of location estimation is heavily dependent on the array geometry and relative source-receiver locations.

Response to Reviewers

We appreciate the time spent by the Reviewers to evaluate our paper. Their comments have been very valuable in helping us to increase the quality of our manuscript by complementing the information and providing new analysis and figures. The modified sentences are highlighted in red colour in the revised version of the paper and supplementary information.

Reviewer #1:

In this contribution, the authors proposed a blind near-field array signal processing method based on fiber-optic distributed acoustic sensor (DAS). DAS has many sensing channels, but not all channels could provide valid and correct information. In view of this, this paper gives a method to evaluate and rank the reliability of each channel without prior knowledge, from which the pilot is selected. Then the pilot trace is improved by delay-and-sum beamforming further. The authors organized contrastive analysis to verify their proposed method with significant results.

The reviewer also have some questions:

1. The proposed method involves lots of calculation. How much time did the authors take for data processing?

Reply: The processing time was about 9 minutes using Matlab in an Intel(R) Core(TM) i7-9750H CPU @ 2.60GHz, as was described in the discussion section of the manuscript. This execution time however corresponds to the worst-case scenario in which all DAS channels are used in the processing. This case was used in the paper only for illustrative purposes, to show how the use of poor-quality channels impairs the array signal processing results. In practice, the computational time can be highly reduced by using a dedicated programming and hardware, as described in the discussion section. We have complemented the description in the lines 419-421 of the revised manuscript.

2.The location of vibration source via DAS is not a new research field. The innovation of the manuscript mainly lies on the improved method for the selection of pilot trace, while the PCCF, time delay of arrival (TDOA), and beamforming methods have already been widely adopted.

Reply: The main novelty of the manuscript lies in the blind methodology proposed to select the most reliable DAS channels for array signal processing and to deal with the nonuniform axial response of DAS sensors. Using no prior information on the acoustic signal nor acoustic source location, the novel approach ranks the measurement reliability of each DAS channel based on the peak-to-root-mean-square ratio of the phase cross-correlation function (PCCF), helping us to blindly identify the most reliable channels and to evaluate distortions in the measured DAS waveforms caused by acoustic reflections. This is information that cannot be obtained if the processing is only based on the PCCF peak. This approach also leads to a sparse array with the most reliable DAS channels being nonuniformly distributed along the sensing fibre. To the best of our knowledge this is also part of the novelty of the manuscript. We have complemented the revised manuscript to highlight some of these aspects, adding text in lines 68-69, 75, 128-129 and 403.

3. The fine processing of the data usually means that the adaptability is reduced. Can the authors prove their method in other scenarios with superior performance?

Reply: In this case, the processing method deals with a challenging scenario, accounting for acoustic reflections, DAS directivity, inhomogeneous propagation media and nonuniform fibre coupling. As noted by the Reviewer, all these adverse conditions impairing DAS measurements inherently reduce the performance of array processing methods. Therefore, a natural improvement in the algorithm results would be expected in scenarios with simpler acoustic field propagation, where DAS measurements are not impaired, as for instance when these adverse conditions are not present and if a uniform response of the sensor array is secured. A comment regarding this has been added to lines 131-133 of the manuscript.

4. In line 494, the author wonders if the Eq. (5) should be Eq. (7).

Reply: We thank the Reviewer for pointing out this typo. The equation is indeed Eq. (7) and not Eq (5). The text has been corrected in line 527 (old line 494) of the revised manuscript.

5. In Ku, Emery M., and Gregory L. Duckworth. "Tracking a human walker with a fiber optic distributed acoustic sensor." Proceedings of Meetings on Acoustics ICA2013. Vol. 19. No. 1. Acoustical Society of America, 2013, the beamform method is utilized in localizing footfalls. So the reviewer wonder whether the method is still useful for the shock signal which lasts a short time like footstep?

Reply: The method can be used for detecting footstep, provided that the DAS sensor has a sharp spatial resolution (e.g., centimetric-scale) and a good temporal sampling to capture the full acoustic bandwidth of the signal. Note however that low-intensity footsteps could be hardly detectable by the optical fibre if this placed hundreds of meters away from the walking area. Nevertheless, this is an issue affecting any DAS sensor and is not specific from the proposed method. The mentioned paper has been included as Ref. 25, and we have discussed it with new text added to the description of the state-of-the-art, in lines 44-46 of the revised manuscript.

Reviewer #2:

Dear Marcelo and Felipe,

It has been a pleasure to read your paper submission, it was very well presented and the processing steps clearly described with interesting results.

The submission demonstrated a novel fully blind process to select high quality DAS channels to 1) improve the signal to noise ratio, and 2) determine accurately the signal source location through beamforming techniques. It was particularly interesting to see the statistics from the analysis of the data sets at the 50 different source locations.

Reply: We thank the Reviewer for the positive comments on our Manuscript.

There are a few points which I'd like to raise.

In figure 3 there are two notable areas 1) between channel 280 and 400 and 2) towards the end of the fiber, channel 800+, where the reliability ranking is poor i.e. very few dark blue dots. Do you have an explanation for why these areas don't show the high levels of reliability ranking as the majority of the installation? This might be the case where the signal is broadside to the cable and therefore the DAS response is reduced? It is difficult to tell without a better understanding of the installation. Can I suggest that figure 1(a) is labeled with channel 0 and ch862?

Reply: We thank the Reviewer for pointing out this interesting feature of the results. Since DAS channels with poor quality measurements are all placed in neighbouring areas (see mentioned channel positions in the upper-right corner of the new Fig. 1(a)), the low ranking of those channels is presumably attributed to the poor strain transfer from the ground to the sensing optical fibre. Note that these channels cover areas with different optical fibre orientations, and therefore the reason mentioned by the Reviewer, i.e., the signal being broadside, could be probably discarded. Based on the Reviewer's suggestion, we have modified Fig. 1(a) by labelling some of the DAS channels to improve understanding, and we have added some comments in lines 189-192 related to the previous description.

The test site had an 'unique' fiber installation pattern. As many of the installed fibers where this technique could be employed i.e. dark telecommunications fibers, are linear installations, what are your thoughts about performance in this, arguably more representative case. In the simplest case, it is worth mentioning that in a linear installation there will be ambiguity in the localization due to the symmetry i.e. which side of the cable is the source.

Reply: We fully agree with the Reviewer on the particularities of the optical fibre layout used in this case, as well as on the ambiguities arising in a linear fibre installation due to the geometric symmetry. However, the use of straight linear fibres may also be inadequate for real-field DAS measurement, due to the directional response of a DAS sensor. In particular, if the acoustic signal is broadside to the cable, no strain will be measured. This is a problem for all DAS sensors and not only for our method. Therefore, robust DAS measurements require the use of an optical fibre installation with different orientations, making the DAS monitoring robust against random directions of the acoustic wave arrival (although some local measurements will still have null or poor response). Incidentally, the use of fibre installations with different orientations also benefits our method to better identify the actual source position without triangulation ambiguity. Following the Reviewer's suggestion, we have added text in lines 388-397 to mention this ambiguity issue when using linear fibre installations and to describe the need of having angular diversity in optical fibre orientations for robust DAS measurements. The requirement of angular diversity has also been added to lines 11 and 76.

Line 25 - I would suggest replacing the phrase 'kHz-to-MHz acoustic bandwidth' with 'wide acoustic bandwidth'. There are quite a few recent publications demonstrating mHz performance of DAS, so best to avoid putting general limits on the DAS acoustic bandwidth.

Reply: We fully agree with the Reviewer on the fact that many recent publications target mHz response, especially for seismic monitoring, and therefore, we have modified the text in line 26 (old line 25) as suggested to avoid any limit on the DAS acoustic bandwidth.

I suggest including the reference below, as far as I know this is the earliest DAS reference to signal localization through beamforming.

Ku, E. M. & Duckworth, G. L. "Tracking a human walker with a fiber optic distributed acoustic sensor." (2013). doi:10.1121/1.4800575.

Reply: We thank the Reviewer for suggesting this article. We have included some text in the description of the state-of-the-art, in lines 44-46, and we have added the paper as reference 25.

Reviewer #3:

The manuscript presents a novel workflow to locate vibration sources using DAS data with imperfect array geometry and dataset. The work is innovative, the results are encouraging, and the discussion is comprehensive. I recommend the publication after minor revisions.

Several claims in the abstract can be confusing and misleading, especially considering most readers read it without knowing the context and method details. For example, Line 11 sounds like the workflow does not need to know the fiber location at all. On line 16, it is confusing to state no optical fiber is required (fiber still has to be close enough). I recommend the authors rephrase these statements.

Reply: We thank the Reviewer for pointing out some potentially confusing and misleading sentences in the abstract. We must however highlight that the text in line 11 is correct and is not misleading because the proposed method for signal enhancement does not need to know the fibre location at all, because the processing to perform the spatial filtering is only based on the DAS measurements and blind estimation of TDOAs. We must however clarify that in this case, due to the blind synchronisation of channels based on

estimated TDOAs, we do not steer the resulting beam toward a pre-defined position, and therefore no spatial information about the sensing fibre is required in this approach. We have clarified this later in the text by adding new comments in lines 428-430 and 433-434 of the modified manuscript. On the other hand, we understand that the sentence in line 16 was possibly misleading. We intended to say that the method allows monitoring the acoustic field emitted in positions far from the optical fibre location. Based on the Reviewer's comment, we modified this last sentence of the abstract in line 16.

It is a bit strange to put the "Method" section at the end. I had difficulties understanding the Result section without first reading the "Method" section. I recommend the authors reorder the sections.

Reply: We appreciate the Reviewer's comment; however, in this case, we followed the conventional structure of Nature Communication scientific articles. As suggested by the author's guidelines for the journal, the main text of the manuscript must describe the novelty and features of the proposed approach, as well as discuss the main results and limitations of the proposal. On the other hand, the Method section, placed after the main text, must describe in detail the techniques used in the work. According to this guideline, we cannot change the position of the Method section.

The beamforming method that the authors proposed is based on the assumption that the source is isotropic, which radiates the same waveform in all directions. This may not be the case in the real world. For example, a shear source radiates waveforms with 180 phase shifts in different directions. It is better to make this assumption clear in the manuscript.

Reply: The Reviewer is partially right. Our general assumption is that the acoustic propagation is isotropic. However, in the anisotropic propagation scenario, the estimated TDOAs can still allow for the proper synchronisation of different phase-shifted DAS channels for signal enhancement, leading to a better representation of the emitted acoustic wave with a given reference phase condition. In the case of source localisation processing, the proposed method works only with isotropic sources since TDOAs are converted into distances, and therefore non-isotropic propagations would induce significant errors in the source localisation. We have commented this by adding lines 108-113 to the revised manuscript.

The workflow that the authors proposed is computationally expensive. I like that the authors admitted and discussed this in the Discussion section. However, I believe more information could be provided. For example, how long the cross-correlation window is, and how many times the current calculation needs to be accelerated (through GPU and parallel computing) in order to achieve real-time analysis, which is critical for this type of application.

Reply: In this case, the cross-correlation window is 30 s, corresponding to 30k data points. In principle, the window should capture the main features of the measured signal to provide reliable estimations of the TDOAs and channel quality. Thus, the window size must be defined depending on the length and spectral content of the signals to be processed. Further research could be still required to optimise the window size for specific applications. On the other hand, determining how many times the current calculation needs to be accelerated for real-time processing highly depends on the target response time required for the specific applications. For instance, if the execution time is reduced to equal the acoustic signal length (30 s in this case), an acceleration factor of 18 would be required. On the other hand, improving the current processing time in a factor 500, for example, would lead to an execution time of about 1 s. Note that these levels of acceleration, compared to our Matlab implementation, can be easily achieved with a proper programming using GPUs. We have added some comments on these two aspects in lines 412-417 and lines 423-427 of the modified manuscript. In addition, we added Ref. 43 to support some statements since this reference provides an exhaustive evaluation of the acceleration factors obtained in cross-correlation calculations using CUDA C programming in GPUs compared to Matlab programming in CPUs.

Other minor comments:

Fig 8: it is good to highlight the channels used for the location estimation in figure b.

Reply: Note that the source location estimation is performed using recursive different numbers of channels and histograms of the estimations, so that the most repeated estimate is chosen. Therefore, there is no optimal number of channels to highlight in a figure. This has been clarified adding new text in lines 299-300. However, we thank the Reviewer for this suggestion, since the idea is interesting to illustrate that the most reliable DAS channels participating in the array processing are arbitrarily distributed along the sensing fibre, leading to a sparse array. Supplementary Note 7 has been added to the Supplementary Information to show the optimal number of 41 channels used by beamforming-based signal enhancement, as well as the distributions of the best 100 and 200 channels used for source localisation. The added information is also cited in the main text, in lines 227-228 and 314-318.

Fig 9: I recommend including a source location error map to better demonstrate the relation between estimation errors and relative positions between source and array.

Reply: We appreciate the Reviewer's suggestion. We added Supplementary Fig. 12 to show the source location error map. The original source positions are shown together to the position estimated by the proposed algorithm for each case. The figure is described in the added Supplementary Note 8 and cited in in lines 367-371 of the main text of the revised manuscript.

Line 378: This is too optimistic. The accuracy of location estimation is heavily dependent on the array geometry and relative source-receiver locations.

Reply: We thank the Reviewer for this comment, since we could realise that the sentence in that mentioned line was too general and optimistic. We have toned-down the sentence by including the requirements for angular diversity in the optical fibre installation and the dependence on the locations of the acoustic source and sensing optical fibre. The new sentences can be found in lines 388 and 497-499 of the revised manuscript.

Remarks to author**(Reviewer #1)**

The authors have addressed my comments and modified the manuscript accordingly, and I have no more questions.

(Reviewer #2)

Thank you for making the changes suggested from my original comments and for your detailed response. I can recommend publication, the work is comprehensively explained with some exciting results.

(Reviewer #3)

The authors have addressed my comments and concerns. I recommend this manuscript be published in its current form.